# Learning Invariants through Soft Unification

**Nuri Cingillioglu**
Imperial College London
nuric@imperial.ac.uk

**Alessandra Russo**
Imperial College London
a.russo@imperial.ac.uk

## Abstract

Human reasoning involves recognising common underlying principles across many examples. The by-products of such reasoning are invariants that capture patterns such as "if someone went somewhere then they are there", expressed using variables "someone" and "somewhere" instead of mentioning specific people or places. Humans learn what variables are and how to use them at a young age. This paper explores whether machines can also learn and use variables solely from examples without requiring human pre-engineering. We propose Unification Networks, an end-to-end differentiable neural network approach capable of lifting examples into invariants and using those invariants to solve a given task. The core characteristic of our architecture is soft unification between examples that enables the network to generalise parts of the input into variables, thereby learning invariants. We evaluate our approach on five datasets to demonstrate that learning invariants captures patterns in the data and can improve performance over baselines.

## 1 Introduction

Humans have the ability to process symbolic knowledge and maintain symbolic thought [46]. When reasoning, humans do not require combinatorial enumeration of examples but instead utilise invariant patterns where specific entities are replaced with placeholders. Symbolic cognitive models [24] embrace this perspective with the human mind seen as an information processing system operating on formal symbols such as reading a stream of tokens in natural language. The language of thought hypothesis [29] frames human thought as a structural construct with varying sub-components such as "X went to Y". By recognising what varies across examples, humans are capable of lifting examples into invariant principles that account for other instances. This symbolic thought with variables is learned at a young age through symbolic play [32]. For instance, a child learns that a sword can be substituted with a stick [15] and engages in pretend play.

Although variables are inherent in symbolic formalisms and their models of computation, as in first-order logic [36], they are pre-engineered and used to solve specific tasks by means of assigning values to them. However, when learning from data only, being able to recognise when and which symbols could act as variables and therefore take on different values is crucial for lifting examples into general principles that are invariant across multiple instances. Figure 1 shows an example invariant learned by our approach: *if someone is the same thing as someone else then they have the same colour*. With this invariant, our approach solves *all* training and test examples in task 16 of the bAbI dataset [48].

*V*:**bernhard** is a *V*:**frog**
*V*:**lily** is a *V*:**frog**
*V*:**lily** is *V*:**green**
───────────────────
what colour is *V*:**bernhard**
───────────────────
green

Figure 1: Invariant learned for bAbI task 16, basic induction, where *V*:**bernhard** denotes a variable with default symbol bernhard. This single invariant accounts for all the training, validation and test examples of this task.

In this paper we address the question of whether a machine can learn and use the notion of a *variable*, i.e. a symbol that can take on different values. For instance, given an example of the form "bernhard is a frog" the machine would learn that the token "bernhard" could be *someone* else and the token "frog" could be *something* else. When the machine learns that a symbol is a variable, assigning a value can be reframed as attending to an appropriate symbol. Attention models [5, 27, 9] allow neural networks to focus, attend to certain parts of the input often for the purpose of selecting a relevant portion. Since attention mechanisms are also differentiable they are often jointly learned within a task. This perspective motivates our idea of a unification mechanism for learning variables across examples that utilises attention and is fully differentiable. We refer to this approach as *soft unification* that can jointly learn which symbols can act as variables and how to assign values to them.

Hence, we propose an end-to-end differentiable neural network approach for learning and utilising the notion of lifting examples into invariants, which can then be used by the network to solve given tasks. The main contribution of this paper is a novel architecture capable of learning and using variables by lifting a given example through soft unification. As a first step, we present the empirical results of our approach in a controlled environment using four synthetic datasets and then with respect to a real-world dataset along with the analysis of the learned invariants that capture the underlying patterns present in the tasks. Our implementation using Chainer [45] is publicly available at `https://github.com/nuric/softuni` with the accompanying data.

## 2   Unification Networks

Reasoning with variables involves first of all identifying what variables are in a given context as well as defining the process by which they are assigned values. The intuition is that when the varying components, i.e. variables, of an example are identified, the example can be lifted into an invariant that captures its structure but with variables replacing its varying components. Such an invariant can account for multiple other instances of the same structure. We present our approach in Algorithm 1 and detail the steps below (note that the algorithm is more compressed than the steps described). Refer to Table 5 in Appendix A for a summary of symbols and notations used in the paper.

**Step 1** (Pick Invariant Example). We start from an example data point to generalise from. If we assume that within a task there is *one* common pattern then any example should be an instance of that pattern. Therefore, we can randomly pick any example within a task from the dataset as our invariant example $G$. In this paper, $G$ is one data point consisting of a context, query and answer (see Table 1).

**Step 2** (Lift Invariant Example). In order for the invariant example to predict other examples correctly, certain symbols might need to *vary* such as **V:bernard** in Figure 1. We capture this degree of *variableness* with a function $\psi : \mathbb{S} \rightarrow [0, 1]$ for every symbol appearing in $G$ where $\mathbb{S}$ is the set of all symbols. When $\psi$ is a learnable function, the model learns to identify the variables and convert the data point into an invariant, i.e. learning invariants. For a certain threshold $\psi(s) \geq t$, we visualise them in bold with a **V** prefix. An invariant is thus a pair $I \triangleq (G, \psi)$, the example data point and the variableness of each symbol. Which symbols emerge as variables depend on whether they need to be assigned new values, but how do we know which values to assign? This brings us to unification.

**Step 3** (Compute Unifying Features). Suppose now we are given a new data point $K$ that we would like to unify with our invariant $I$ from the previous step. $K$ might start with a question "what colour is lily" and our invariant "what colour is **V:bernard**". We would like to match bernard with lily. However, if we were to just use $d$-dimensional representations of symbols $\phi : \mathbb{S} \rightarrow \mathbb{R}^d$, the representations of bernard and lily would need to be similar which might confound an upstream predictor network, e.g. when lily and bernard appear in the same story it would be difficult to distinguish them with similar representations. To resolve this issue, we learn *unifying features* $\phi_U : \mathbb{S} \rightarrow \mathbb{R}^d$ that intuitively capture some common meaning of two otherwise distinct symbols. For example, $\phi(\text{bernard})$ and $\phi(\text{lily})$ could represent specific people whereas $\phi_U(\text{bernard}) \approx \phi_U(\text{lily})$ the notion of *someone*; similarly, in Figure 3, $\phi_U(7) \approx \phi_U(3)$ the notion of the head of a sequence. Notice how we use the original symbol bernard in computing the representation of **V:bernard**; this is intended to capture the variable's bound *meaning* following the idea of referants [14].

**Step 4** (Soft Unification). Using learned unifying features, for every variable in $I$ we can now find a corresponding symbol in $K$. This process of unification, i.e. replacing variables with symbol values, is captured by the function $g$ that given an invariant and another example it updates the variables with appropriate values. To achieve this, we compute a soft attention for each symbol $s$ in the invariant

**Algorithm 1:** Unification Networks

**Input:** Invariant $I$ consisting of example $G$ and variableness network $\psi$, example $K$, features network $\phi$, unifying features network $\phi_U$, upstream predictor network $f$

**Output:** Predicted label for example $K$

```
1  begin ▷ Unification Network
2  │   return f ∘ g(I, K)                         ▷ Predictions using Soft Unification g
3  begin ▷ Soft Unification function g
4  │   foreach symbol s in G do
5  │   │   A_{s,:} ← φ(s)                          ▷ Features of G, A ∈ ℝ^{|G|×d}
6  │   │   B_{s,:} ← φ_U(s)                         ▷ Unifying features of G, B ∈ ℝ^{|G|×d}
7  │   foreach symbol s in K do
8  │   │   C_{s,:} ← φ(s)                          ▷ Features of K, C ∈ ℝ^{|K|×d}
9  │   │   D_{s,:} ← φ_U(s)                         ▷ Unifying features of K, D ∈ ℝ^{|K|×d}
10 │   Let P = softmax(BD^T)                       ▷ Attention map over symbols, P ∈ ℝ^{|G|×|K|}
11 │   Let E = PC                                  ▷ Attended representations of G, E ∈ ℝ^{|G|×d}
12 │   foreach symbol s in G do
13 │   │   U_{s,:} ← ψ(s)E_{s,:} + (1 − ψ(s))A_{s,:}   ▷ Unified representation of I, U ∈ ℝ^{|G|×d}
14 │   return U
```

using the unifying features $\phi_U(s)$ (line 10 in Algorithm 1) and interpolate between its own and its variable value (line 13 in Algorithm 1). Since $g$ is a differentiable formulation and $\psi$, $\phi$ and $\phi_U$ can be learnable functions, we refer to this step as *soft unification*. In Figure 3, the variable *V:7* is changed towards the symbol 3, having learnt that the unifiable feature is the head of the sequence.

**Step 5** (Predict). So far we have constructed a unified data point of $I$ with the new example $K$ of which we would like to predict the answer. How do we predict? We use another, potentially upstream task specific, network $f$ that tries to predict the answer based on our unified input. Recall that our data points are triples of the form context, query and answer. In question answering, $f$ could be a memory network, or when working with grid like inputs, a CNN. By predicting on the output of our unification, we expect that $f \circ g(I, K) = f(K) = a$. If $f$ is differentiable, we can learn how to unify while solving the upstream task. We focus on $g$ and use standard networks for $f$ to understand which invariants are learned and the interaction of $f \circ g$ instead of the raw performance of $f$.

## 3 Instances of Unification Networks

We present four architectures to model $f \circ g$ and demonstrate the flexibility of our approach towards different architectures and upstream tasks. Except in Unification RNN, the $d$-dimensional representation of symbols are learnable embeddings $\phi(s) = O[s]^T E$ with $E \in \mathbb{R}^{|\mathbb{S}|×d}$ randomly initialised by $\mathcal{N}(0, 1)$ and $O[s]$ the one-hot encoding of the symbol. The variableness of symbols are learnable weights $\psi(s) = \sigma(w_s)$ where $w \in \mathbb{R}^{|\mathbb{S}|}$ and $\sigma$ is the sigmoid function. We consider every symbol independently as a variable irrespective of its surrounding context and leave further contextualised formulations as future work. However, unifying features $\phi_U$ can be context sensitive to disambiguate same symbol variables appearing in different contexts. Full details the of models, including hyper-parameters, are available in Appendix A.

**Unification MLP (UMLP)** ($f$: MLP, $g$: RNN) We start with a sequence of symbols as input, e.g. a sequence of digits 4234. Unifying features $\phi_U$, from Step 3, are obtained using the hidden states of a bi-directional GRU [10] processing the embedded sequences. In Step 5, the upstream MLP predicts the answer based on the flattened representation of the unified sequence.

**Unification CNN (UCNN)** ($f$: CNN, $g$: CNN) To adapt our approach for a grid of symbols, we use *separate* convolutional neural networks with the same architecture to compute unifying features $\phi_U$ as well as to predict the correct answer through $f$ in Step 5. We mask out padding in Step 4 to avoid assigning null values to variables.

**Unification RNN (URNN)** ($f$: RNN, $g$: MLP) We start with a varying length sequence of words such as a movie review. We set ConceptNet word embeddings [43] as $\phi$ to compute $\phi_U(s) = \text{MLP}(\phi(s))$ and $\psi(s) = \sigma(\text{MLP}(\phi(s)))$. Then, the final hidden state of $f$, an LSTM [20], predicts the answer.

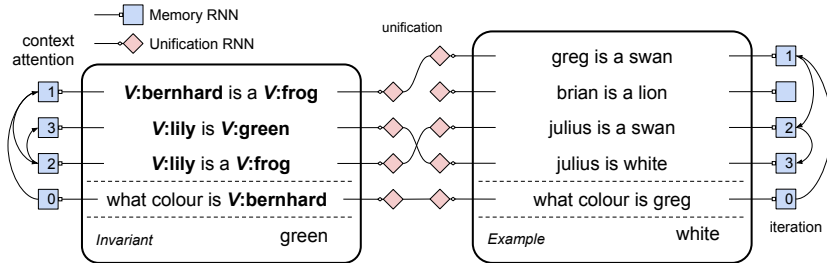

Figure 2: Graphical overview of soft unification within a memory network (UMN). Each sentence is processed by two bi-directional RNNs for memory and unification. At each iteration the context attention selects which sentences to unify and the invariant produces the same answer as the example.

**Unification Memory Networks (UMN)** ($f$: MemNN, $g$: RNN) Soft unification does not need to happen prior to $f$ in a $f \circ g$ fashion but can also be incorporated at any intermediate stage multiple times. To demonstrate this ability, we unify the symbols at different memory locations at each iteration of a Memory Network [47]. We take a list of lists as input such as a tokenised story, Figure 2. The memory network $f$ uses the final hidden state of a bi-directional GRU (blue squares in Figure 2) as the sentence representations to compute a context attention, i.e. select the next context sentence starting with the query. With the sentences attended to, we can unify the words of the sentences at each iteration following Steps 2 to 4. We use another bi-directional GRU (pink diamonds in Figure 2) for unifying features $\phi_U$. Following line 13 in Algorithm 1, the new unified representation of the memory slot (the sentence) is used by $f$ to perform the next iteration. The prediction is then based on the final hidden state of the invariant example. This setup, however, requires pre-training $f$ such that the context attentions match the correct pairs of sentences to unify which limits the performance of the combined network by how well $f$ performs.

Although we assume a single pattern in Step 1, a task might contain slightly different examples such as "Where is X?" and "Why did X go to Y?". To let the models potentially learn and benefit from different invariants, we can pick multiple examples to generalise from and aggregate the predictions from each invariant. One simple approach is to sum the predictions of the invariants $\sum_{I \in \mathbb{I}} f \circ g(I, K)$ used in UMLP, UCNN and URNN where $\mathbb{I}$ is the set of invariants. For UMN, at each iteration we weigh the hidden states from each invariant using a bilinear attention $\eta = \text{softmax}(h_I^0 W h_K^{0^T})$ where $h_I^0$ and $h_K^0$ are the representations of the query (at iteration 0).

Table 1: Sample context, query and answer triples and their training sizes *per task*. For the distribution of generated number of examples per task on Sequence and Grid data refer to Appendix B.

| Dataset | Context | Query | Answer | Training Size |
|---|---|---|---|---|
| Sequence | 8384 | duplicate | 8 | $\leq$ 1k, $\leq$ 50 |
| Grid | 0 0 3<br>0 1 6<br>8 5 7 | corner | 7 | $\leq$ 1k, $\leq$ 50 |
| bAbI | Mary went to the kitchen.<br>Sandra journeyed to the garden. | Where is Mary? | kitchen | 1k, 50 |
| Logic | p(X) ← q(X).<br>q(a). | p(a). | True | 2k, 100 |
| Sentiment A. | easily one of the best films | Sentiment | Positive | 1k, 50 |

## 4 Datasets

We use five datasets consisting of context, query and an answer $(C, Q, a)$ (see Table 1 and Appendix B for further details) with varying input structures: fixed or varying length sequences, grids and nested sequences (e.g. stories). In each case we use an appropriate model: UMLP for fixed length sequences, UCNN for grid, URNN for varying length sequences and UMN for iterative reasoning.

**Fixed Length Sequences** We generate sequences of length $l = 4$ from 8 unique symbols represented as digits to predict (i) a constant, (ii) the head of the sequence, (iii) the tail and (iv) the duplicate symbol. We randomly generate 1000 triples and then only take the unique ones to ensure the test split

contains unseen examples. The training is then performed over a 5-fold cross-validation. Figure 3 demonstrates how the invariant '*V:7* 4' can predict the head of another example sequence '3 9'.

**Grid** To spatially organise symbols, we generate a grid of size $3 \times 3$ from 8 unique symbols. The grids contain one of (i) $2 \times 2$ box of identical symbol, (ii) a vertical, diagonal or horizontal sequence of length 3, (iii) a cross or a plus shape and (iv) a triangle. In each task we predict (i) the identical symbol, (ii) the head of the sequence, (iii) the centre of the cross or plus and (iv) the corner of the triangle respectively. We generate 1000 triples discarding any duplicates.

**bAbI** The bAbI dataset consists of 20 synthetically generated natural language reasoning tasks (refer to [48] for task details). We take the 1k English set and use 0.1 of the training set as validation. Each token is lower cased and considered a unique symbol. Following previous works [39, 44], we take multiple word answers also to be a unique symbol. To initially form the repository of invariants, we use the bag-of-words representation of the questions and find the most dissimilar ones based on their cosine similarity as a heuristic to obtain varied examples.

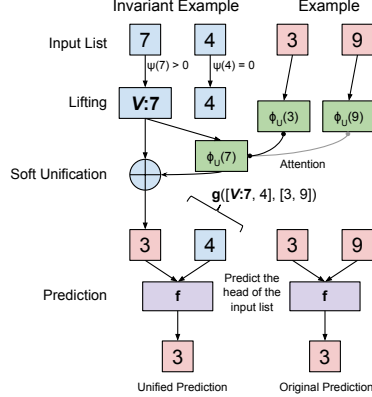

Figure 3: Graphical overview of predicting the head of a sequence using an invariant and soft unification where $g$ outputs the new sequence 3 4.

**Logical Reasoning** To distinguish our notion of a variable from that used in logic-based formalisms, we generate logical reasoning tasks in the form of logic programs using the procedure from [11]. The tasks involve learning $f(C, Q) =$ True if and only if $C \vdash Q$ over 12 classes of logic programs exhibiting varying paradigms of logical reasoning including negation by failure [12]. We generate 1k and 50 logic programs per task for training with 0.1 as validation and another 1k for testing. Each logic program has one positive and one negative prediction giving a total of 2k and 100 data points respectively. We use one random character from the English alphabet for predicates *and* constants, e.g. $p(p, p)$ and an upper case character for logical variables, e.g. $p(X, Y)$. Further configurations such as restricting the arity of predicates to 1 are presented in Table 11 Appendix D.

**Sentiment Analysis** To evaluate on a noisy real-world dataset, we take the sentiment analysis task from [41] and prune sentences to a maximum length of 20 words. We threshold the scores $\leq 0.1$ and $\geq 0.9$ for negative and positive labels respectively to ensure unification cannot yield a neutral score, i.e. the model is forced to learn either a positive or a negative label. We then take 1000 or 50 training examples *per label* and use the remaining $\approx 676$ data points as unseen test examples.

## 5  Experiments

We probe three aspects of soft unification: the impact of unification on performance over unseen data, the effect of multiple invariants and data efficiency. To that end, we train UMLP, UCNN and URNN with and without unification and UMN with pre-training using 1 or 3 invariants over either the entire training set or only 50 examples. Every model is trained via back-propagation using Adam [22] with learning rate 0.001 on an Intel Core i7-6700 CPU using the following objective function:

$$J = \overbrace{\lambda_K \mathcal{L}_{\text{nll}}(f)}^{\text{Original output}} + \lambda_I [ \; \overbrace{\mathcal{L}_{\text{nll}}(f \circ g)}^{\text{Unification output}} + \tau \overbrace{\sum_{s \in \mathbb{S}} \psi(s)}^{\text{Sparsity}} ] \qquad (1)$$

where $\mathcal{L}_{\text{nll}}$ is the negative log-likelihood with sparsity regularisation over $\psi$ at $\tau = 0.1$ to discourage the models from utilising spurious number of variables. We add the sparsity constraint over the variableness of symbols $\psi(s)$ to avoid the trivial solution in which every symbol is a variable and $G$ is completely replaced by $K$ still allowing $f$ to predict correctly. Hence, we would like the *minimal* transformation of $G$ towards $K$ to expose the common underlying pattern. For UMLP and UCNN, we set $\lambda_K = 0$, $\lambda_I = 1$ for training just the unified output and the converse for the non-unifying versions. For URNN, we set $\lambda_K = \lambda_I = 1$ to train the unified output and set $\lambda_I = 0$ for non-unifying version.

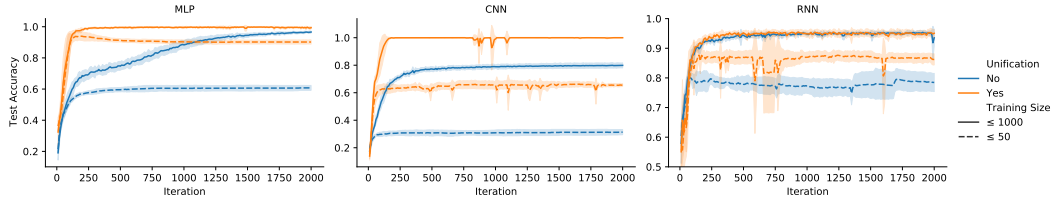

Figure 4: Test accuracy over iterations for UMLP, UCNN and URNN models with 1 invariant versus no unification. Soft unification aids with data efficiency of models against their plain counterparts.

To pre-train the UMN, we start with $\lambda_K = 1$, $\lambda_I = 0$ for 40 epochs then set $\lambda_I = 1$ to jointly train the unified output. For UMN, we also add the mean squared error between hidden states of $I$ and $K$ at each iteration (see Appendix C). In the strongly supervised cases, the negative log-likelihood of the context attentions (which sentences are selected at each iteration) are also added. Further training details including sample training curves are available in Appendix C.

Figure 4 portrays how soft unification generalises better to unseen examples in test sets over plain models. Despite $f \circ g$ having more trainable parameters than $f$ alone, this data efficiency is visible across all models when trained with only $\leq 50$ examples *per task*. We believe soft unification architecturally biases the models towards learning unifying features that are common across examples, therefore, potentially also common to unseen examples. The data efficient nature is more emphasised with UMLP and UCNN on synthetic datasets in which there are unambiguous patterns in the tasks and they achieve higher accuracies in as few as 250 iterations (batch updates) against their plain counterparts. In the real-world dataset of sentiment analysis, we observe a less steady training curve for URNN and performance as good as if not better than its plain version. The fluctuations in accuracy around iterations 750 to 1000 in UCNN and iteration 700 in URNN are caused by penalising $\psi$ which forces the model to adjust the invariant to use less variables half way through training. Results with multiple invariants are identical and the models learn to ignore the extra invariants (Figure 11 Appendix D) due to the regularisation applied on $\psi$ zeroing out unnecessary invariants. Training with different learning rates overall paint a similar picture (Figure 8 Appendix C).

Table 2: Aggregate error rates (%) on bAbI 1k for UMN and baselines N2N [44], GN2N [26], EntNet [19], QRN [39] and MemNN [47] respectively. Full comparison is available in Appendix D.

| Training Size | | 1k | | | 50 | | | 1k | | | |
|---|---|---|---|---|---|---|---|---|---|---|---|
| Supervision | Weak | | Strong | | | | Weak | | | | Strong |
| # Invs / Model | 1 | 3 | 1 | 3 | 3 | N2N | GN2N | EntNet | QRN | MemNN |
| Mean | 18.8 | 19.0 | **5.1** | 6.6 | 28.7 | 13.9 | 12.7 | 29.6 | 11.3 | 6.7 |
| # > 5% | 10 | 9 | 3 | 3 | 17 | 11 | 10 | 15 | 5 | 4 |

For iterative reasoning tasks, Tables 2 and 3 aggregate the results for our approach against comparable baseline memory networks which are selected based on whether they are built on Memory Networks (MemNN) [47] and predict by iteratively updating a hidden state. For example, End-to-End Memory Networks (N2N) [44] and Iterative Memory Attention (IMA) [11] networks update a hidden state vector after each iteration by attending to a single context sentence similar to our architecture. We observe that strong supervision and more data per task yield lower error rates which is consistent with previous work reflecting how $f \circ g$ can be bounded by the efficacy of $f$ modelled as a memory network. In a weak supervision setting, i.e. when sentence selection is not supervised, our model attempts to unify arbitrary sentences often failing to follow the iterative reasoning chain. As a result,

Table 3: Aggregate task error rates (%) on the logical reasoning dataset for UMN and baseline IMA [11]. Individual task results are available in Appendix D.

| Model | | | UMN | | | | IMA | |
|---|---|---|---|---|---|---|---|---|
| Training Size | | | 2k | | | 100 | 2k | |
| Supervision | Weak | | Strong | | | | Weak | Strong |
| # Invs | 1 | 3 | 1 | 3 | 3 | | - | |
| Mean | 37.7 | 37.6 | **27.4** | 29.0 | 47.1 | | 38.8 | 31.5 |
| # > 5% | 10 | 10 | 10 | 11 | 12 | | 11 | 11 |

only in the supervised case we observe a minor improvement over MemNN by 1.6 in Table 2 and over IMA by 4.1 in Table 3. Without reducing the performance of $f$, our approach is still able to learn invariants as shown in Figure 6. This dependency on $f$ also limits the ability of $f \circ g$ to learn from 50 and 100 examples per task failing 17/20 of bAbI and 12/12 of logical reasoning tasks respectively. The increase in error rate with 3 invariants in Table 3, we speculate, stems from having more parameters and more pathways, rendering training more difficult and slower.

Table 4: Number of *exact* matches of the learnt invariants with expected ones in synthetic datasets where the invariant is known. If the model does not get an exact match, it might use more or fewer variables than expected while still solving the task. For sequences and grid datasets, there are 5 folds each with 4 tasks giving 20 and the logic dataset has 12 tasks with 3 runs giving 36 invariants.

| Model | UMLP | | UCNN | | UMN | |
|---|---|---|---|---|---|---|
| Dataset | Sequences | | Grid | | Logic | |
| Train Size | $\leq 1k$ | $\leq 50$ | $\leq 1k$ | $\leq 50$ | 2k | |
| Supervision | - | | | | Weak | Strong |
| Correct / Total | 18/20 | 18/20 | 13/20 | 14/20 | 7/36 | 23/36 |
| Accuracy (%) | 90.0 | 90.0 | 65.0 | 70.0 | 19.4 | 63.9 |

For synthetic sequences, grid and logic datasets in which we know exactly what the invariants *can be*, Table 4 shows how often our approach captures *exactly* the expected invariant. We threshold $\psi$ as explained in Section 6 and check for an exact match; for example for predicting the head of a sequence in UMLP, we compare the learnt invariant against the pattern "$V$_ _ _". Although with increasing dataset complexity the accuracy drops, it is important to note that just because the model does not capture the exact invariant it may still solve the task. In these cases, it may use extra or more interestingly fewer variables as further discussed in Section 6.

# 6 Analysis

Figure 5 shows an invariant for sentiment analysis in which words such as silly that contribute more to the sentiment have a higher $\psi$. Intuitively, if one replaces 'silly' with the adjective 'great', the sentiment will change. The replacement, however, is not a hard value assignment but an interpolation (line 13 Algorithm 1) which may produce a new intermediate representation from $G$ towards $K$ different enough to allow $f$ to predict correctly. Since we penalise the magnitude of $\psi$ in equation 1, we expect these values to be as low as possible. For synthetic datasets, we apply a threshold $\psi(s) > t$ to extract the learned invariants and set $t$ to be the mean of the variable symbols as a heuristic except for bAbI where we use $t = 0.1$. The magnitude of $t$ depends on the amount of regularisation $\tau$, equation 1, number of iterations and batch size. Sample invariants shown in Figure 6 describe the patterns present

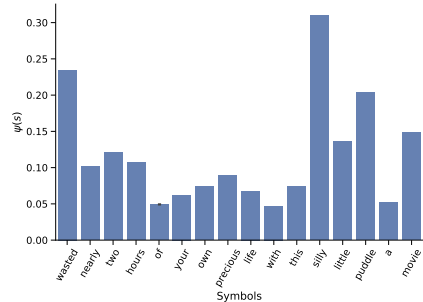

Figure 5: The variableness $\psi(s)$ of an invariant for sentiment analysis with words such as 'silly' emerging as variables.

in the tasks with parts that contribute towards the final answer becoming variables. Extra symbols such as 'is' or 'travelled' do not emerge as variables, as shown in Figure 6a; we attribute this behaviour to the fact that changing the token 'travelled' to 'went' does not influence the prediction but changing the action, the value of $V$:**left** to 'picked' does. However, based on random initialisation, our approach can convert an arbitrary symbol into a variable and let $f$ compensate for the unifications it produces. For example, the invariant "$V$:**8** 5 2 2" could predict the tail of another example by unifying the head with the tail using $\phi_U$ of those symbols in Step 3. Further examples are shown in Appendix D. Pre-training $f$ as done in UMN seems to produce more robust and consistent invariants since, we speculate, a pre-trained $f$ encourages more $g(I, K) \approx K$.

**Interpretability versus Ability** A desired property of interpretable models is transparency [25]. A novel outcome of the learned invariants in our approach is that they provide an approximation of the underlying general principle that may be present in the data. Figure 6e captures the structure of multi-hop reasoning in which a predicate $V$:**n** can entail another $V$:**i** with a matching constant $V$:**o** if there is a chain of rules that connect the two predicates. However, certain aspects regarding the ability

|  |  |
|---|---|
| **V:john** travelled to the **V:office** | this **V:morning V:bill** went to the **V:school** |
| **V:john V:left** the **V:football** | yesterday **V:bill** journeyed to the **V:park** |
| where is the **V:football** | where was **V:bill** before the **V:school** |
| office | park |

(a) bAbI task 2, two supporting facts. The model also learns **V:left** since people can also drop or pick up objects potentially affecting the answer.

(b) bAbI task 14, time reasoning. **V:bill** and **V:school** are recognised as variables alongside **V:morning** capturing *when* someone went.

| | | |
|---|---|---|
| 5 8 6 4 const 2 | 0 **V V**   0 1 0   0 0 1 | **V:i** ( T ) ← **V:l** ( T ), |
| **V:8** 3 3 1 head 8 | 0 **V V**   6 **V** 8   0 5 4 | **V:l** ( U ) ← **V:x** ( U ), |
| 8 3 1 **V:5** tail 5 | 0 0 0   0 7 0   7 8 **V** | **V:x** ( K ) ← **V:n** ( K ), |
| **V:1** 4 3 **V:1** dup 1 | box   centre   corner | **V:n** ( **V:o** ) ⊢ **V:i** ( **V:o** ) |

(c) Successful invariants learned with UMLP using 50 training examples only shown as $(C, Q, a)$.

(d) Successful invariants learned with UCNN. Variable default symbols are omitted for clarity.

(e) Logical reasoning task 5 with arity 1. The model captures how **V:n** could entail **V:i** in a chain.

Figure 6: Invariants learned across the four datasets using the three architectures. For iterative reasoning datasets, bAbI and logical reasoning, they are taken from strongly supervised UMN.

of the model such as how it performs temporal reasoning, are still hidden inside $f$. In Figure 6b, although we observe **V:morning** as a variable, the overall learned invariant captures nothing about how changing the value of **V:morning** alters the behaviour of $f$, i.e. how $f$ uses the interpolated representations produced by $g$. The upstream model $f$ might look *before* or *after* a certain time point **V:bill** went somewhere depending what **V:morning** binds to. Without the regularising term on $\psi(s)$, we initially noticed the models using, what one might call extra, symbols as variables and binding them to the same value occasionally producing unifications such as "bathroom bathroom to the bathroom" and still $f$ predicting, unsurprisingly, the correct answer as bathroom. Hence, regularising $\psi$ with the correct amount $\tau$ in equation 1 to reduce the capacity of unification seems critical in extracting not just any invariant but one that represents the common structure.

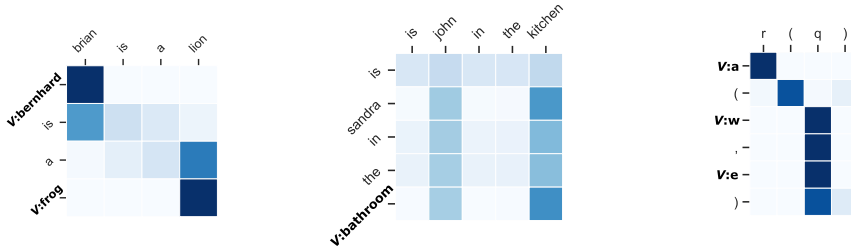

(a) bAbI task 16. A one-to-one mapping is created between variables **V:bernhard** with brian and **V:frog** with lion.

(b) bAbI task 6. **V:bathroom** is recognised as the only variable creating a one-to-many binding to capture the same information.

(c) Logical reasoning task 1. An arity 2 predicate is forced to bind with arity 1 creating a many-to-one binding.

Figure 7: Variable bindings from line 10 in Algorithm 1. Darker cells indicate higher attention values.

Attention maps from line 10 in Algorithm 1 reveal three main patterns: one-to-one, one-to-many or many-to-one bindings as shown in Figure 7 with more in Appendix D. Figure 7a captures what one might expect unification to look like where variables unify with their corresponding counterparts, e.g. **V:bernhard** with brian and **V:frog** with lion. However, occasionally the model can optimise to use less variables and *squeeze* the required information into a single variable, for example by binding **V:bathroom** to john and kitchen as shown in Figure 7b. We believe this occurs due to the sparsity constraint on $\psi(s)$ encouraging the model to be as conservative as possible. Since the upstream network $f$ is also trained, it has the capacity to compensate for condensed or malformed unified representations; a possible option could be to freeze the upstream network while learning to unify. Finally, if there are more variables than needed as in Figure 7c, we observe a many-to-one binding with **V:w** and **V:e** mapping to the same constant q. This behaviour begs the question how does the model differentiate between $p(q)$ and $p(q, q)$. We speculate the model uses the magnitude of $\psi(w) = 0.037$ and $\psi(e) = 0.042$ to encode the difference despite both variables unifying with the same constant.

# 7 Related Work

Learning an underlying general principle in the form of an invariant is often the means for arguing for generalisation in neural networks. For example, Neural Turing Machines [16] are tested on previously unseen sequences to support the view that the model might have captured the underlying pattern or algorithm. In fact, [47] claim "MemNNs can discover simple linguistic patterns based on verbal forms such as (X, dropped, Y), (X, took, Y) or (X, journeyed to, Y) and can successfully generalise the meaning of their instantiations." However, this claim is based on the output of $f$ and unfortunately it is unknown whether the model has truly learned such a representation or indeed is utilising it. Our approach sheds light on this ambiguity and presents these linguistic patterns explicitly as invariants, ensuring their utility through $g$ without solely analysing the output of $f$ on previously unseen symbols. Although we associate these invariants with our existing understanding of the task to perhaps mistakenly anthropomorphise the machine, for example by thinking it has learned *V:mary* as *someone*, it is important to acknowledge that these are just symbolic patterns. They do not make our model, in particular $f$, more interpretable in terms of how these invariants are used or what they mean to the model. In these cases, our interpretations may not necessarily correspond to any understanding of the machine, relating to the Chinese room argument [37].

Learning invariants by lifting ground examples is related to least common generalisation [33] by which inductive inference is performed on facts [40] such as generalising *went(mary,kitchen)* and *went(john,garden)* to *went(X,Y)*. Unlike in a predicate logic setting, our approach allows for soft alignment and therefore generalisation between varying length sequences. Existing neuro-symbolic systems [7] focus on inducing rules that adhere to *given* logical semantics of what variables and rules are. For example, $\delta ILP$ [13] constructs a network by rigidly following the given semantics of first-order logic. Similarly, Lifted Relational Neural Networks [42] ground first-order logic rules into a neural network while Neural Theorem Provers [34] build neural networks using backward-chaining [36] on a given background knowledge base with templates. However, the notion of a variable is pre-defined rather than learned with a focus on presenting a practical approach to solving certain problems, whereas our motivation stems from a cognitive perspective.

At first it may seem the learned invariants, Section 6, make the model more interpretable; however, this transparency is not of the model $f$ but of the data. The invariant captures patterns that potentially approximates the data generating distribution but we still do not know *how* the model $f$ uses them upstream. Thus, from the perspective of explainable artificial intelligence (XAI) [1], learning invariants or interpreting them does not constitute an explanation of the reasoning model $f$ even though "if *someone* goes *somewhere* then they are there" might look like one. Instead, it can be perceived as causal attribution [28] in which someone being somewhere is attributed to them going there. This perspective also relates to gradient based model explanation methods such as Layer-Wise Relevance Propagation [4] and Grad-CAM [38, 8]. Consequently, a possible view on $\psi$, Section 2, is a gradient based usefulness measure such that a symbol utilised upstream by $f$ to determine the answer becomes a variable similar to how a group of pixels in an image contribute more to its classification. However, gradient based saliency methods have shown to be unreliable if based solely on visual assessment [2].

Finally, one can argue that our model maintains a form of counterfactual thinking [35] in which soft unification $g$ creates counterfactuals on the invariant example to alter the output of $f$ towards the desired answer, Step 5. The question *where Mary would have been if Mary had gone to the garden instead of the kitchen* is the process by which an invariant is learned through multiple examples during training. This view relates to methods of causal inference [31, 21] in which counterfactuals are vital as demonstrated in structured models [30].

# 8 Conclusion

We presented a new approach for learning variables and lifting examples into invariants through the usage of soft unification. Application of our approach to five datasets demonstrates that Unification Networks perform comparatively if not better to existing architectures without soft unification while having the benefit of lifting examples into invariants that capture underlying patterns present in the tasks. Since our approach is end-to-end differentiable, we plan to apply this technique to multi-modal tasks in order to yield multi-modal invariants for example in visual question answering.

**Acknowledgements**

We would like to thank Murray Shanahan for his helpful comments, critical feedback and insights regarding this work. We also thank Anna Hadjitofi for proof-reading and improving clarity throughout the writing of the paper.

## Broader Impact

As it is with any machine learning model aimed at extracting patterns solely from data, learning invariants through soft unification is prone to being influenced by spurious correlations and biases that might be present in the data. There is no guarantee that even a clear, high accuracy invariant might correspond to a valid inference or casual relationship as discussed in Section 6 with some mis-matching invariants presented in Appendix D. As a result, if our approach succeeds in solving the task with an invariant, it does not mean that there is only pattern or in the case of failing to do so, a lack of patterns in the data. There has been recent work [3, 18] on tackling a different notion of invariance formed of features that are consistent (hence invariant) across different training dataset environments, to learn more robust predictors. Our method is instead targeted at research and researchers involved with combining cognitive aspects such as variable learning and assignment with neural networks under the umbrella of neuro-symbolic systems [7, 6]. A differentiable formulation of variables could accelerate the research of combining logic based symbolic systems with neural networks. In summary, we regard this work as an experimental stepping stone towards better neuro-symbolic systems in the domain of artificial intelligence research.

**Funding Disclosure:** This work is not supported by any direct or indirect third-party funding.

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
