[Supplementary Material]

# A Model Details

This appendix section describes each instance of the Unification Networks from Section 3 in more detail. In all cases, we select the hyper-parameters such as number of layers and embedding dimensions based on similar previous work.

Table 5: A non-exhaustive list of symbols and notation used throughout the paper and their descriptions. Note that the notation in the Appendix sections might differ.

| Notation | Description |
|---|---|
| $G$ | Invariant example data point of which the variable symbols are learnt. It is derived from the term **g**round example that will be lifted. |
| $(C, Q, a)$ | A single data point consisting of a context, query and an answer. |
| $\psi$ | Variableness function that given a symbol outputs a value between 0 and 1. |
| $\mathbb{S}$ | Set of all symbols appearing in a dataset or given example data point. |
| $I$ | Invariant that consists of an example $G$ and variableness function $\psi$. |
| $\mathbb{I}$ | Set of invariants when multiple invariants are used. |
| $\boldsymbol{V}$ | Variable symbol prefix, used to indicate symbols that are variables, i.e. have $\psi(s) \geq t$ above a certain threshold $t$. |
| $\boldsymbol{V\text{:bernhard}}$ | Variable with default symbol bernhard, bernhard is the $s$ in $\psi(s) \geq t$. |
| $\phi$ | Features of a symbol, it could be any d-dimensional representation $\phi(s) \in \mathbb{R}^d$. |
| $\phi_U$ | Unifying features of a symbol, $\phi_U(s) \in \mathbb{R}^d$. |
| $K$ | A new example data point of which we want to the predict the answer. |
| $g$ | Soft unification function that computes the unified representation of $I$, defined in Algorithm 1. |
| $f$ | Upstream, potentially task specific, predictor network. |
| $\mathcal{L}_{\text{nll}}$ | The negative log-likelihood loss. |

## A.1 Unification MLP

The input example is a sequence of symbols with a fixed length $l$, e.g. a sequence of digits 4234. Given an embedded input $\phi(K) \in \mathbb{R}^{l \times d}$, the upstream MLP computes the output symbol based on the flattened representations $f(\phi(K)) = \text{softmax}(\boldsymbol{h}\boldsymbol{E}^T)$ where $\boldsymbol{h} \in \mathbb{R}^d$ is the output of the last layer and $\boldsymbol{E}$ is the embedding matrix for the symbols. However, to compute the unifying features $\phi_U$, $g$ uses a bi-directional GRU [10] running over $\phi(K)$ such that $\phi_U(K) = \Phi(K)\boldsymbol{W}_U$ where $\Phi(K) \in \mathbb{R}^{l \times d}$ is the hidden state of the GRU at every symbol in $K$ and $\boldsymbol{W}_U \in \mathbb{R}^{d \times d}$ is a learnable weight.

To model $f$ as a multi-layer perceptron, we take symbol embeddings of size $d = 16$ and flatten sequences of length $l = 4$ into an input vector of size $\phi(\boldsymbol{k}) \in \mathbb{R}^{64}$. The MLP consists of 2 hidden layers with $\tanh$ non-linearity of sizes $2d$ and $d$ respectively and an output layer of size $d$. To process the query, we concatenate the one-hot encoding of the task id to $\phi(\boldsymbol{k})$ yielding a final input of size $64 + 4 = 68$. For unification features $\phi_U$, we use a bi-directional GRU with hidden size $d$ and the corresponding task id as the initial state. In this case we embed the task ids using another learnable embedding matrix. The hidden state at each symbol is taken with a linear transformation to give $\phi_U(s) = \boldsymbol{W}_U \Phi(s)$ where $\Phi(s)$ is the hidden state of the bi-directional GRU. The variable assignment is then computed as an attention over the context sentence the according to Algorithm 1.

## A.2 Unification CNN

Given a grid of embedded symbols $\phi(K) \in \mathbb{R}^{w \times h \times d}$ where $w$ is the width and $h$ the height, we use a convolutional neural network such that the final prediction is $f(\phi(K)) = \text{softmax}((\boldsymbol{W}\boldsymbol{h} + \boldsymbol{b})\boldsymbol{E}^T)$

where $\boldsymbol{h}$ this time is the result of global max pooling and $\boldsymbol{W}, \boldsymbol{b}$ are learnable parameters. We also model $g$ using a *separate* convolutional network with the same architecture as $f$ and set $\phi_U(K) = c_2(\mathrm{relu}(c_1(K)))$ where $c_1, c_2$ are the convolutional layers. The grid is padded with 0s to obtain $w \times h \times d$ after each convolution such that every symbol has a unifying feature. This model conveys how soft unification can be adapted to the specifics of the domain, for example by using a convolution in a spatially structured input.

We take symbols embeddings of size $d = 32$ to obtain an input grid $\phi(\boldsymbol{K}) \in \mathbb{R}^{3 \times 3 \times 32}$. Similar to UMLP, for each symbol we append the task id as a one-hot vector to get an input of shape $3 \times 3 \times (32 + 4)$. Then $f$ consists of 2 convolutional layers with $d$ filters each, kernel size of 3 and stride 1. We use relu non-linearity in between the layers. We pad the grid with 2 columns and 2 rows to a $5 \times 5$ such that the output of the convolutions yield again a hidden output $\mathsf{H} \in \mathbb{R}^{3 \times 3 \times d}$ of the same shape. As the final hidden output $h$, we take a global max pool to over $\mathsf{H}$ to obtain $\boldsymbol{h} \in \mathbb{R}^d$. Unification function $g$ is modelled identical to $f$ without the max pooling such that $\phi_U(\boldsymbol{K}_{ij}) = \mathsf{H}'_{ij}$ where $\mathsf{H}'$ is the hidden output of the convolutional layers.

### A.3 Unification RNN

The input example is a variable length sequence of words such that $\phi(K) \in \mathbb{R}^{l \times d}$ where $l$ is the length of the sequence and $d$ is the embedding size. The upstream $f$ is modelled as an LSTM [20] which processes the input sequence to yield a final hidden state $\boldsymbol{h} \in \mathbb{R}^d$ such that $\boldsymbol{h} = \mathrm{LSTM}(\phi(K))$. To obtain the final prediction, we apply a linear transformation followed by the sigmoid non-linearity, $f(\phi(K)) = \sigma(\boldsymbol{W}_f \boldsymbol{h} + b_f)$ where $\boldsymbol{W}_f \in \mathbb{R}^{1 \times d}$ and $b_f$ are learnable parameters. To compute the symbol features $\phi$, their unifying features $\phi_U$ and their variableness $\psi$, we use a linear transformation on the original word embedding of a symbol $\boldsymbol{e}_s$ such that $\phi(s) = \boldsymbol{W} \boldsymbol{e}_s + \boldsymbol{b}$, $\phi_U(s) = \boldsymbol{W}\phi(s) + \boldsymbol{b}$ and $\psi(s) = \sigma(\boldsymbol{W}\phi(s) + b)$ where all $\boldsymbol{W}$ and $\boldsymbol{b}$ are *distinct* learnable parameters.

In the sentiment analysis dataset, Section 4, we start with ConceptNet word embeddings [43] for each symbol $\boldsymbol{e}_s \in \mathbb{R}^{300}$. The input words are then projected down to $d = 16$ dimensions to compute the required features above. We apply a dropout of 0.5 to $\boldsymbol{h}$ and let the LSTM skip padding added to shorter sentences. The initial states of the LSTM are set as zero vectors.

### A.4 Unification Memory Networks

The memory network $f$ uses the final hidden state of a bi-directional GRU (blue squares in Figure 2) as the sentence representations to compute a context attention. At each iteration, we unify the words between the attended sentences using the same approach in UMLP with another bi-directional GRU (pink diamonds in Figure 2) for unifying features $\phi_U(\text{bernhard}) = \boldsymbol{W}_U \Phi(\text{bernhard})$. Following line 13 in Algorithm 1, the new unified representation of the memory slot is computed and $f$ uses it to perform the next iteration. Concretely, $g$ produces an unification tensor $\mathsf{U} \in \mathbb{R}^{M \times m \times N \times d}$ where $M$ and $m$ is the number of sentences and words in the invariant respectively, and $N$ is the number of sentences in the example such that after the context attentions are applied over $M$ and $N$, we obtain $\phi(\boldsymbol{k}) \in \mathbb{R}^{m \times d}$ as the unified sentence with variables instantiated at that iteration. In other words, we compute all pairwise sentence unification and then use the context attentions from the invariant and the example to reduce the unification tensor $\mathsf{U}$. Note that unlike in the UMLP case, the sentences can be of varying length. The prediction is then $\mathrm{softmax}(\boldsymbol{W}\boldsymbol{h}_I^J + \boldsymbol{b})$ where $\boldsymbol{h}_I^J$ is the hidden state of the memory network running over the invariant after $J$ iterations. This setup, however, requires pre-training $f$ such that the context attentions match the correct pairs of sentences to unify.

Unlike previous architectures, with UMN we interleave $g$ into $f$. We use embedding sizes of $d = 32$ and model $f$ with an iterative memory network. We take the final hidden state of a bi-directional GRU, with initial state $\boldsymbol{0}$, $\Phi_M$ to represent the sentences of the context $\boldsymbol{C}$ and query $\boldsymbol{q}$ in a $d$-dimensional vector $\boldsymbol{M}_i = \Phi_M(\boldsymbol{C}_i)$ and the query $\boldsymbol{m}_q = \Phi_M(\boldsymbol{q})$. The initial state of the memory network is $\boldsymbol{h}^0 = \boldsymbol{m}_q$. At each iteration $j$:

$$\boldsymbol{A}_i^j = \tanh(\boldsymbol{W}\rho(\boldsymbol{M}_i, \boldsymbol{h}^j) + \boldsymbol{b}) \tag{2}$$

$$\beta^j = \mathrm{softmax}(\boldsymbol{W}\Phi_A(\boldsymbol{A}^j) + \boldsymbol{b}) \tag{3}$$

where $\Phi_A$ is another $d$-dimensional bi-directional GRU and $\rho(\boldsymbol{x}, \boldsymbol{y}) = [\boldsymbol{x}; \boldsymbol{y}; \boldsymbol{x} \odot \boldsymbol{y}; (\boldsymbol{x} - \boldsymbol{y})^2]$ with $\odot$ the element-wise multiplication and $[;]$ the concatenation of vectors. Taking $\beta^j$ as the context

attention, we obtain the next state of the memory network:

$$h^{j+1} = \sum_i \beta_i^j \tanh(W \rho(M_i, h^j) + b) \qquad (4)$$

and iterate $J$ many times in advance. The final prediction becomes $f(C, q) = \text{softmax}(W h^J + b)$. All weight matrices $W$ and bias vectors $b$ are independent but are tied across iterations. The intuition is that $A^j$ captures some interaction between the current state and the memory slots. Then the bi-directional GRU is used to model the temporal relationship, e.g. if there are multiple sentences where Mary went, we would like to select the last location. Finally, another interaction is computed between each memory slot and the current state to obtain what the state would be if that memory slot was selected. Using $\beta^j$, the state is weighed based on which memory slot was selected at that iteration.

# B   Generated Dataset Samples

Table 6: Sample context, query and answer triples from sequences and grid tasks.

| Dataset | Task | Context | Query | Answer |
|---------|------|---------|-------|--------|
| Sequence | i | 1488 | constant | 2 |
| Sequence | ii | 6157 | head | 6 |
| Sequence | iii | 1837 | tail | 7 |
| Sequence | iv | 3563 | duplicate | 3 |
| Grid | i | 0 0 0<br>0 2 2<br>8 2 2 | box | 2 |
| Grid | ii | 4 0 0<br>0 7 0<br>8 0 1 | head | 4 |
| Grid | iii | 0 6 0<br>1 7 2<br>0 3 0 | centre | 7 |
| Grid | iv | 8 0 0<br>5 6 0<br>2 4 1 | corner | 2 |

Table 7: Training sizes for randomly generated fixed length sequences and grid tasks with 8 unique symbols. The reason for Grid task (i) to be smaller is because there are at most 32 combinations of $2 \times 2$ boxes in a $3 \times 3$ grid with 8 unique symbols. The upper and lower bounds are taken from the 5-folds that are generated and might differ with each new random test split of the data.

| Task | Sequences | Grid |
|------|-----------|------|
| i | $704.7 \pm 12.8$ | $25.6 \pm 1.8$ |
| ii | $709.4 \pm 13.8$ | $623.7 \pm 14.1$ |
| iii | $709.7 \pm 14.0$ | $768.2 \pm 12.5$ |
| iv | $624.8 \pm 12.4$ | $795.2 \pm 10.3$ |

## C   Training Details

### C.1   Unification MLP & CNN & RNN

Both unification models are trained on a 5-fold cross-validation over the generated datasets for 2000 iterations with a batch size of 64. In this context, each iteration is a single batch update. We don't use any weight decay and save the training and test accuracies every 10 iterations, as presented in Figure 4. We also present the training curves with different learning rates in Figure 8.

### C.2   Unification Memory Networks

We again use a batch size of 64 and pre-train $f$ for 40 epochs then $f$ together with $g$ for 260 epochs. We use epochs for UMN since the dataset sizes are fixed. To learn $g$ alongside $f$, we combine error signals from the unification of the invariant and the example. The objective function not only incorporates the negative log-likelihood $\mathcal{L}_{\mathrm{nll}}$ of the answer but also the mean squared error between intermediate states $\boldsymbol{h}_I^j$ and $\boldsymbol{h}_K^j$ at each iteration as an auxiliary loss:

$$J = \mathcal{L}_{\mathrm{nll}}(f(K), a) + \lambda_I \left[ \mathcal{L}_{\mathrm{nll}}(f \circ g(I,K), a) + \frac{1}{J} \sum_{j=0}^{J}(\boldsymbol{h}_I^j - \boldsymbol{h}_K^j)^2 + \tau \sum_{s \in \mathbb{S}} \psi(s) \right] \quad (5)$$

We pre-train by setting $\lambda_I = 0$ for 40 epochs and then set $\lambda_I = 1$. For strong supervision we also compute the negative log-likelihood $\mathcal{L}_{\mathrm{nll}}$ for the context attention $\beta^j$, described in Appendix A, at each iteration using the supporting facts of the tasks. We apply a dropout of 0.1 for all recurrent neural networks used and only for the bAbI dataset weight decay with 0.001 as the coefficient.

### C.3   Training Curves

Figure 8: Results of Unification MLP, CNN and RNN with different levels of learning rates. We observe that improvement is maintained except with 0.01 with which our approach degrades in training performance. With the more real-world dataset, we can see that our approach maintains a noisier (fluctuating) training curve.

Figure 9: Training curves for UMN on the bAbI dataset with strong supervision, 1 invariant and 1k training size. Note the initial 40 epochs are for pre-training the memory network and then unification is enabled which causes the sudden jump in the training curves. After unification is enabled, our approach learns to solve the tasks through soft unification within 5 epochs for most tasks. Tasks involving positional reasoning and path finding still prove difficult emphasising the dependency of our approach on the upstream memory network $f$ which struggles to solve or over-fits.

Figure 10: Training curves for UMN on the logic dataset with strong supervision, 1 invariant and 2k training size. Note the initial 40 epochs are for pre-training the memory network and then unification is enabled. We observe that besides the very basic tasks that involve just Facts or Unification (these are single iteration tasks), our approach tends to over-fit.

# D   Further Results

Figure 11: Results of Unification MLP, CNN and RNN on increasing number of invariants with different learning rates. There is no impact on performance when more invariants per task are given. We speculate that the models converge to using 1 invariant because the datasets can be solved with a single invariant due to the regularisation applied on $\psi$ zeroing out unnecessary invariants. This behaviour highlights the level of abstraction we can achieve through $\psi$. One might consider "2 *V* 7 *V*" and "*V* 8 4 *V*" to be different patterns or invariants but at a higher level of abstraction they can both represent the concept of a repeated symbol irrespective of the position of the repeating item. Thus, extra invariants can be ignored in favour of one that accounts for different examples through the learnable unifying features $\phi_U$, which might look for the notion of a *repeated symbol* regardless of its position in the sequence. Given the capacity to learn unifying features and the pressure to use the least amount of variables, the models in these cases can optimise to learn and use a single invariant.

Table 8: Individual task error rates on bAbI tasks for Unification Memory Networks. Following previous work, these are the best error rates out of 3 runs and are obtained by taking the test accuracy at the epoch for which the validation accuracy is highest. For the variance across runs, you can refer to the sample training curves for each task in Figure 9.

| Supervision | Weak | | Strong | | |
|---|---|---|---|---|---|
| # Invs | 1 | 3 | 1 | 3 | 3 |
| Training Size | 1k | 1k | 1k | 1k | 50 |
| 1 | 0.0 | 0.0 | 0.0 | 0.0 | 1.1 |
| 2 | 62.3 | 60.4 | 0.1 | 0.4 | 40.4 |
| 3 | 58.8 | 63.7 | 1.2 | 1.3 | 52.1 |
| 4 | 0.0 | 0.0 | 0.0 | 0.0 | 36.9 |
| 5 | 1.9 | 1.6 | 0.5 | 1.6 | 29.7 |
| 6 | 0.0 | 0.1 | 0.0 | 0.0 | 15.4 |
| 7 | 20.5 | 22.3 | 6.4 | 7.7 | 22.4 |
| 8 | 7.4 | 7.7 | 4.2 | 2.9 | 31.9 |
| 9 | 0.3 | 0.0 | 0.0 | 0.0 | 20.6 |
| 10 | 0.1 | 0.5 | 0.2 | 0.3 | 26.5 |
| 11 | 0.0 | 0.0 | 0.0 | 0.0 | 21.1 |
| 12 | 0.0 | 0.0 | 0.0 | 0.0 | 23.5 |
| 13 | 0.4 | 4.7 | 0.0 | 0.2 | 5.6 |
| 14 | 15.3 | 17.4 | 0.1 | 0.1 | 57.3 |
| 15 | 17.8 | 0.0 | 0.0 | 0.0 | 0.0 |
| 16 | 52.7 | 53.3 | 0.0 | 0.0 | 45.4 |
| 17 | 39.9 | 49.3 | 49.5 | 48.4 | 45.8 |
| 18 | 7.2 | 7.9 | 0.3 | 0.8 | 10.9 |
| 19 | 90.4 | 90.7 | 38.9 | 67.8 | 86.2 |
| 20 | 0.0 | 0.0 | 0.0 | 0.0 | 1.8 |
| Mean | 18.8 | 19.0 | **5.1** | 6.6 | 28.7 |
| Std | 27.7 | 28.0 | 13.6 | 18.0 | 21.8 |
| # > 5% | 10 | 9 | 3 | 3 | 17 |

Table 9: Comparison of individual task error rates (%) on the bAbI [48] dataset of the best run. We preferred 1k results if a model had experiments published on both 1k and 10k for data efficiency. We present our approach (UMN) with a single invariant for weak and strong supervision, taken from Table 8. References from left to right: [44] - [26] - [19] - [39] - Ours - [49] - [17] - [47] - Ours - [23]

| Support Size Model | | | Weak | | | | | | Strong | |
| --- | --- | --- | --- | --- | --- | --- | --- | --- | --- | --- |
| | | | 1k | | | 10k | | 1k | | 10k |
| Model | N2N | GN2N | EntNet | QRN | UMN | DMN+ | DNC | MemNN | UMN | DMN |
| 1 | 0.0 | 0.0 | 0.7 | 0.0 | 0.0 | 0.0 | 0.0 | 0.0 | 0.0 | 0.0 |
| 2 | 8.3 | 8.1 | 56.4 | 0.5 | 62.3 | 0.3 | 0.4 | 0.0 | 0.1 | 1.8 |
| 3 | 40.3 | 38.8 | 69.7 | 1.2 | 58.8 | 1.1 | 1.8 | 0.0 | 1.2 | 4.8 |
| 4 | 2.8 | 0.4 | 1.4 | 0.7 | 0.0 | 0.0 | 0.0 | 0.0 | 0.0 | 0.0 |
| 5 | 13.1 | 1.0 | 4.6 | 1.2 | 1.9 | 0.5 | 0.8 | 2.0 | 0.5 | 0.7 |
| 6 | 7.6 | 8.4 | 30.0 | 1.2 | 0.0 | 0.0 | 0.0 | 0.0 | 0.0 | 0.0 |
| 7 | 17.3 | 17.8 | 22.3 | 9.4 | 20.5 | 2.4 | 0.6 | 15.0 | 6.4 | 3.1 |
| 8 | 10.0 | 12.5 | 19.2 | 3.7 | 7.4 | 0.0 | 0.3 | 9.0 | 4.2 | 3.5 |
| 9 | 13.2 | 10.7 | 31.5 | 0.0 | 0.3 | 0.0 | 0.2 | 0.0 | 0.0 | 0.0 |
| 10 | 15.1 | 16.5 | 15.6 | 0.0 | 0.1 | 0.0 | 0.2 | 2.0 | 0.2 | 2.5 |
| 11 | 0.9 | 0.0 | 8.0 | 0.0 | 0.0 | 0.0 | 0.0 | 0.0 | 0.0 | 0.1 |
| 12 | 0.2 | 0.0 | 0.8 | 0.0 | 0.0 | 0.0 | 0.0 | 0.0 | 0.0 | 0.0 |
| 13 | 0.4 | 0.0 | 9.0 | 0.3 | 0.4 | 0.0 | 0.1 | 0.0 | 0.0 | 0.2 |
| 14 | 1.7 | 1.2 | 62.9 | 3.8 | 15.3 | 0.2 | 0.4 | 1.0 | 0.1 | 0.0 |
| 15 | 0.0 | 0.0 | 57.8 | 0.0 | 17.8 | 0.0 | 0.0 | 0.0 | 0.0 | 0.0 |
| 16 | 1.3 | 0.1 | 53.2 | 53.4 | 52.7 | 45.3 | 55.1 | 0.0 | 0.0 | 0.6 |
| 17 | 51.0 | 41.7 | 46.4 | 51.8 | 39.9 | 4.2 | 12.0 | 35.0 | 49.5 | 40.6 |
| 18 | 11.1 | 9.2 | 8.8 | 8.8 | 7.2 | 2.1 | 0.8 | 5.0 | 0.3 | 4.7 |
| 19 | 82.8 | 88.5 | 90.4 | 90.7 | 90.4 | 0.0 | 3.9 | 64.0 | 38.9 | 65.5 |
| 20 | 0.0 | 0.0 | 2.6 | 0.3 | 0.0 | 0.0 | 0.0 | 0.0 | 0.0 | 0.0 |
| Mean | 13.9 | 12.7 | 29.6 | 11.3 | 18.8 | **2.8** | 3.8 | 6.7 | 5.1 | 6.4 |
| # > 5% | 11 | 10 | 15 | 5 | 10 | 1 | 2 | 4 | 3 | 2 |

Table 10: Individual task error rates (%) of UMN against baseline IMA [11] with 2k training size. Following previous work in Table 2, we present the best task error rates across 3 runs. Each error rate is obtained by taking the test accuracy at the epoch when validation accuracy is highest. For the variance across runs, you can refer to the sample training curves for each task in Figure 10.

| Model | | UMN | | | | | IMA | |
| --- | --- | --- | --- | --- | --- | --- | --- | --- |
| Training Size | | 2k | | | | 100 | 2k | |
| Supervision | | Weak | | Strong | | | Weak | Strong |
| # Invs | 1 | 3 | 1 | 3 | 3 | | - | |
| Facts | 1.6 | 0.9 | 0.2 | 0.3 | 37.0 | | 2.8 | 3.8 |
| Unification | 2.4 | 3.0 | 2.1 | 8.8 | 45.0 | | 9.7 | 7.2 |
| 1 Step | 48.9 | 45.2 | 21.1 | 18.7 | 49.2 | | 42.6 | 36.3 |
| 2 Steps | 49.7 | 49.1 | 37.6 | 36.9 | 48.4 | | 49.5 | 36.7 |
| 3 Steps | 48.9 | 49.9 | 35.4 | 36.9 | 46.6 | | 48.3 | 35.9 |
| AND | 35.1 | 35.6 | 16.4 | 18.9 | 48.7 | | 40.3 | 21.4 |
| OR | 37.2 | 38.1 | 32.3 | 32.9 | 45.9 | | 38.4 | 25.3 |
| Transitivity | 50.0 | 50.0 | 25.5 | 23.2 | 49.4 | | 48.6 | 49.3 |
| 1 Step NBF | 30.8 | 30.8 | 19.6 | 23.2 | 46.8 | | 39.1 | 34.2 |
| 2 Steps NBF | 49.9 | 50.0 | 38.8 | 48.0 | 49.0 | | 49.5 | 34.1 |
| AND NBF | 48.5 | 48.6 | 49.8 | 49.7 | 49.4 | | 48.8 | 45.5 |
| OR NBF | 49.2 | 50.4 | 49.8 | 50.0 | 49.7 | | 48.4 | 48.8 |
| Mean | 37.7 | 37.6 | **27.4** | 29.0 | 47.1 | | 38.8 | 31.5 |
| # > 5% | 10 | 10 | 10 | 11 | 12 | | 11 | 11 |

Table 11: Individual task error rates (%) of UMN with extra configurations on the logical reasoning dataset as well as the reported results of baselines DMN and IMA from [11]. Note that the baselines results are trained on a different instance of the dataset that uses the same data generating script. When the logic programs are restricted to arity 1, the length of the predicates become fixed, e.g. $p(a)$, which simplifies soft unification by removing any ambiguity. In this case, the identity matrix would produce the correct value assignments for variables in our approach and as a result we see a clear improvement over arity 2 version of the dataset. Note that one cannot generate transitivity tasks with only arity 1 predicates. Generating more logic programs improves the overall performance of our apporach which fails to solve only 5 tasks with strong supervision and a single invariant.

| Size | 2k | | | | 20k | | | | | 100 | 40k | |
|---|---|---|---|---|---|---|---|---|---|---|---|---|
| Support | Weak | | Strong | | Weak | | Strong | | | | Weak | |
| Arity | 1 | 2 | 1 | 2 | 1 | 2 | 1 | 2 | 2 | 2 | 2 | 2 |
| # Invs / Model | 1 | 3 | 1 | 3 | 1 | 3 | 1 | 1 | 3 | 3 | DMN | IMA |
| Facts | 1.2 | 0.9 | 0.0 | 0.4 | 0.0 | 0.0 | 0.0 | 0.0 | 0.0 | 33.5 | 0.0 | 0.0 |
| Unification | 0.0 | 10.3 | 0.0 | 10.8 | 0.0 | 0.0 | 0.0 | 0.0 | 0.0 | 41.3 | 13.0 | 10.0 |
| 1 Step | 50.3 | 49.8 | 4.4 | 20.0 | 1.2 | 27.8 | 0.1 | 1.3 | 5.7 | 50.2 | 26.0 | 2.0 |
| 2 Steps | 47.5 | 50.0 | 5.7 | 35.0 | 37.2 | 47.8 | 0.0 | 29.7 | 28.7 | 49.9 | 33.0 | 5.0 |
| 3 Steps | 47.6 | 49.2 | 10.4 | 38.7 | 39.6 | 45.6 | 0.0 | 26.0 | 26.1 | 48.3 | 23.0 | 6.0 |
| AND | 31.3 | 37.4 | 10.7 | 16.4 | 29.8 | 29.0 | 0.2 | 0.4 | 1.2 | 50.0 | 20.0 | 5.0 |
| OR | 25.2 | 38.1 | 21.0 | 35.0 | 20.5 | 30.2 | 4.4 | 20.6 | 17.4 | 47.6 | 13.0 | 3.0 |
| Transitivity | | 50.0 | | 26.6 | | 39.6 | | 5.0 | 6.0 | 49.2 | 50.0 | 50.0 |
| 1 Step NBF | 46.4 | 38.7 | 3.8 | 28.8 | 1.1 | 21.6 | 0.1 | 1.1 | 8.0 | 47.6 | 21.0 | 2.0 |
| 2 Steps NBF | 48.5 | 48.9 | 7.7 | 39.6 | 30.4 | 48.2 | 0.1 | 33.4 | 28.7 | 50.3 | 15.0 | 4.0 |
| AND NBF | 51.0 | 50.1 | 43.1 | 48.6 | 29.4 | 44.2 | 0.1 | 1.3 | 40.1 | 49.5 | 16.0 | 8.0 |
| OR NBF | 51.4 | 48.4 | 50.8 | 47.3 | 47.6 | 47.8 | 21.3 | 27.6 | 30.5 | 47.3 | 25.0 | 14.0 |
| Mean | 36.4 | 39.3 | 14.3 | 28.9 | 21.5 | 31.8 | **2.4** | 12.2 | 16.0 | 47.1 | 21.2 | 9.1 |
| Std | 18.7 | 15.9 | 16.4 | 14.1 | 17.1 | 16.7 | 6.1 | 13.2 | 13.6 | 4.7 | 12.3 | 13.4 |
| # > 5% | 9 | 11 | 7 | 11 | 7 | 10 | 1 | 5 | 9 | 12 | 11 | 5 |

---

**V:sandra** went back to the **V:bathroom**

is **V:sandra** in the **V:bathroom**

yes

Figure 12: bAbI task 6, yes or no questions. The invariant captures the varying components of the story which are the person and the location they have been, similar to task 1 of the bAbI dataset. However, the invariant captures nothing about how this task is actually solved by the upstream network $f$, i.e. how $f$ uses the interpolated unified representation.

$$V{:}m\,(\,V{:}e\,) \quad \vdash V{:}m\,(\,V{:}e\,)$$
$$V{:}a\,(\,V{:}w\,,\,V{:}e\,) \quad \vdash V{:}a\,(\,V{:}w\,,\,V{:}e\,)$$
$$V{:}m\,(\,T\,) \quad \vdash V{:}m\,(\,c\,)$$
$$V{:}x\,(\,A\,) \leftarrow not\ V{:}q\,(\,A\,) \quad \vdash V{:}x\,(\,V{:}z\,)$$

Figure 13: Invariants learned on tasks 1, 2 and 11 with arity 1 and 2 from the logical reasoning dataset. Last invariant on task 11 lifts the example around the negation by failure, denoted as *not*, capturing its semantics. We present the context on the left and the query on the right, $C \vdash q$ means $f(C, q) = 1$ as described in Section 4.

3 **V:4** 7 **V:8** head
7 4 **V:2** **V:6** head
**V:4** 3 1 **V:5** tail
**V:3** **V:3** 5 **V:6** duplicate

(a) Invariants with extra variables learned with UMLP.

| **V:4** 0 | 0 | 0 **V:4** 0 | 6 | 4 **V:3** |
|---|---|---|---|---|
| 0 1 | 0 | 1 **V:6** 8 | 0 **V:2** | 8 |
| 0 0 **V:3** | 0 | 2 0 | 0 0 | 7 |
| head | | centre | corner | |

(b) Mismatching invariants learned with UCNN.

Figure 14: Invariants learned that do not match the data generating distribution from UMLP and UCNN using ≤ 1000 examples to train. In these instances the unification still binds to the the correct symbols in order to predict the desired answer, i.e. quantitatively we get the same results. With these invariants, either the upstream network $f$ or the unifying features $\phi_U$ compensate for the extra or mis-matching variables.

(a) bAbI task 2. When a variable is unused in the next iteration, e.g. **V:football**, it unifies with random tokens often biased by position.

(b) Logical reasoning task 1. A one-to-one alignment is created between predicates and constants.

(c) Logical reasoning task 3. Arity 1 atom forced to bind with arity 2 creates a one-to-many mapping.

Figure 15: Further attention maps from line 10 in Algorithm 1, darker cells indicate higher attention values.