[Reviews · NeurIPS 2020]

Review 1

Summary and Contributions: The paper proposes Unification Networks that perform soft unification between examples to identify variables in the input and thus learn invariants. Instantiations of the framework are described for four data types -- 1D sequence of symbols, 2D grid of symbols, word sequences of varying length, and lists of lists (e.g., tokenized stories). Models are evaluated on five datasets and the results both quantitatively and qualitatively presented and analyzed.

Strengths: * Novel framework for learning invariants * Results show that the proposed method is effective * Paper is well-written

Weaknesses: * Evaluation is somewhat weak as in more baselines should be introduced and compared with * Baseline methods like N2N and GN2N are not described in the paper * Figure 2 is hard to follow

Correctness: Yes

Clarity: Yes

Relation to Prior Work: Yes, but description of some baselines are missing. It would also help to clearly discuss differences of the proposed method with respect to previous works and baselines.

Reproducibility: Yes

Additional Feedback: Please see weaknesses


Review 2

Summary and Contributions: This paper presents a framework for logical reasoning via learning variables/invariants in given examples, and then using them to be substituted with values to make logical deductions. The key idea is to use a soft attention on every symbol in an example to find if that symbol is a variable or not. The paper then presents "unification networks" that unifies the identified variables with their values from the test time example, to produce the deductions. Experiments are provided on tasks with varying complexity and demonstrate some promise.

Strengths: The idea of of identifying invariants through soft-attention in expressions via task-specific examples in the way the paper is placed, is perhaps novel as far as I know, and is useful in various problems.

Weaknesses: While, the overarching idea is interesting, I think the paper lacks technical insights or experimental support to substantiate its claims. Specifically, 1. I think the main weakness is that there is no quantitative analysis measuring the performance of the proposed framework in correctly identifying (or grounding) the invariants in the examples. There are a few qualitative results in Figures 6 and 7, however, I think reporting this invariant classification accuracy is important, as that determines if the network is essentially using the proposed idea or is it something else that leads to improvements. To this end, Figure 5 reports this analysis on the sentiment dataset, however, the attention weights are low (max is 0.3), suggesting that the proposed system is not very confident in learning these invariants? Of course, this dataset is perhaps noisy, which is why I think it is better to report the numbers on the other cleaner datasets. 2. It is not clear to me why the paper resorted to a soft-interpolation in (1)? Won't such a soft-interpolation produce varying symbol embeddings? That is, if \phi(s) and \phi(k) are two symbols, won't using (1-\psi(s))\phi(s) + \psi(s) \phi(k) lead to a new symbol embedding that is neither of \phi(s) and \psi(k), such that the original semantic meaning of either symbol is lost? Won't that hurt the interpretability of the attention as it percolates to other predicates in the examples? 3. How will the soft-attention resolve ambiguities in the examples? For example, suppose given "X is a Y", "R is a S", and a few other predicates on X,Y, R, and S. Suppose the test example is A is a B, won't the soft-interpolation lead to an ambiguous embedding for A as it can relate to both X and R? and so for B? I believe there is must be a context embedding that could help here, but that is not provided in the soft-interpolation formulation in (1)? Or am I missing something?

Correctness: Please see above.

Clarity: The paper is clearly written and easy to read.

Relation to Prior Work: I think the related work could be improved via more exact comparisons to prior works. For example, the paper argues that Neural Turing Machines may not be interpretable as much as the proposed framework, as the attention on the symbols provide a better grounding and more interpretable results. This is also similar to arguments in favor or Grad-CAM and related approaches. However, the extend of their interpretability has often been questioned; for example [a] below. It would be good if the authors can comment on this, their view of such issues, and how they could provide more convincing analysis of their results. [a] Sanity Checks for Saliency Masks, Adebayo et al., NeurIPS 2018

Reproducibility: Yes

Additional Feedback:


Review 3

Summary and Contributions: ################## update: Thank you for the clarifications! I decided to update my score to a 6, based on the response and discussion. However, I only make my 6 conditional on the fact that the authors will run and include the analysis and ablations on the synthetic dataset, otherwise, I would recommend resubmission instead. I would like to point out that my question about LR was not answered in the authors' response. By looking at the code it looks like only one value (1e-3) was used for all experiments. Given that experiments are quite simple, I encourage the authors to rerun them with more LRs, especially for RNNs it can make a huge difference and it would make results more robust overall. ########################## The authors explore how to make a model learn invariances, by discovering variables from the data in an end-to-end differentiable way. Through a series of 5 steps, the model learns to identify and reuse the knowledge acquired across examples. In step (1) a random example is chosen, in step (2) a neural net (psi) identifies the variables, in step (3) another neural network computes unifying features (phi_U) which map different values to the (approx) same latent; in step (4) the variables found in the original input are used to identify their match in the new input (using soft attention on phi_U); step (5) makes a prediction with one more neural network f.

Strengths: I found the empirical evaluation generally convincing, with several datasets and using different “modes”. It’s also good that the authors tested this on several architectures (MLPs, RNNs, CNNs). Significance: learning invariances is a key component of generalisation. I think the approach taken in the paper is interesting and very relevant to the community. Novelty: I am not very familiar with works that explicitly aim at learning variables and at using them to make predictions, but reading the related work section, the proposed approach appears novel to me.

Weaknesses: The empirical evaluation would benefit for more clarity regarding the experiments: what hyper parameters were used? How were they selected? I could not find any of these info on the paper (nor in the appendix). Of course results on higher dimensional datasets would be interesting to see, as most of the datasets used here are more “conceptual”, but I think this is not absolutely necessary. Neural networks are notoriously tricky to optimise, especially when they involve RNNs as it is the case here. Even small variations of the learning rate can result in models with drastically different performances, so for example running an experiment with a fixed (and unknown) lr decreases the confidence in how general the results are. Without an algorithm box, I had trouble understanding the exact method based exclusively on the text descriptions of sections 2 and 3, and Figures 2 and 3. While the authors shared the code (which I didn’t run but looks very clean!), so I expect the results would be reproduced, as of now it would be hard for a reader to build on top of this work only based on the paper itself, and one would probably have to reverse-engineer the algorithm from the code.

Correctness: They appear to be correct, but seeing hyperparameters and training curves would help build confidence in the soundness of the results.

Clarity: I enjoyed reading the abstract and the introduction, they were clear and set a good motivation for the following part of the paper. I struggled more with section 2, which I had to read three times, and I am still not sure I have a very clear grasp of all the components. I went to the appendix as well but it did not clarify things completely. I had trouble keeping up with all of phi’s forms (phi, phi_I, phi_U, phi_K), and I am still not sure whether phi and phi_U are different models or not, and how g is related to phi_U. Figure 2 helped a little, but I still couldn’t grasp the exact number of models, their roles, and how they interact: I think an algorithm box is really essential to have a clear understanding of the proposed method.

Relation to Prior Work: The related work section is rich, but does not go into details about e.g. the difference between the proposed approach and similar works. Which of the papers discussed is most similar to this? How do they differ?

Reproducibility: No

Additional Feedback: Unfortunately without understanding the details of the proposed method (due to a lack of an algorithm box or pseudocode) it is hard to make a recommendation of acceptance. I liked the paper’s general motivation, and I hope the authors can provide an algorithm box in their rebuttal for the reviewers to get a better understanding. I opted for a score of 5 due to the important weaknesses in the current version, but I will revise my score after reading the rebuttal. As of now I like the paper at a high level but I feel in its current form it is not ready to be published as it would be difficult to know what *exactly* the authors proposed and build upon it. - Other questions: How many examples are used at once to identify invariances? Just one? - Very small fixes: On Preview at least it is not possible to click on labels (e.g. of figures, tables, refs) and have the pdf directed to the corresponding object. Maybe it was not exported correctly to pdf? Table 3 has a weird horizontal shift of the vertical line next to the rightmost “Weak”


Review 4

Summary and Contributions: The paper proposes a soft unification approach that can be used to introduce elements of symbolic reasoning to diverse neural models. Briefly, for a given symbolic query K, one takes another example G and replaces certain symbols in G with certain symbols in K. The results of such substitution g(G, K) is used for the downstream processing alongside with the original example K. The paper demonstrates how the proposed approach can be used with 4 different downstream models and how it leads to better performance in low data settings.

Strengths: I think the idea is quite novel and presented quite well. Both qualitative and quantitative results are clearly presented. The topic of learning something akin to symbolic reasoning should be of high interest to NeurIPS attendants.

Weaknesses: As it is often the case with conceptually elegant approaches, as the difficulty of the task increases, the benefits brought by the proposed approach look less convincing.

Correctness: The paper is mostly sound, but I do have some concerns. In Table 3, which reports logical reasoning results, it would be good to see a more apples-to-apples comparison. I don’t see a baseline without soft unification. The results for arity of 1 and 2 are reported side by side: does it make sense? Comparison to the baselines from the literature is difficult because a different number of examples is used. Very small \psi(s) magnitudes of around 0.03 are interpreted as the model having selected the symbol s as a variable. I am wondering if such interpretation is correct.

Clarity: The paper is, in general, well written, but it’s very very dense. Section 2 was a very difficult, dense read at the first attempt. At Step 3 I was completely lost. Figure 2 is introduced before the respective task is explained. Several notation symbols were not properly introduced: \mathcal{S}, \mathcal{G}, \mathcal{I}. The baselines from the literature are not presented very clearly. A reminder of what is “strong supervision” and “weak supervision” would be needed to make the paper more self-contained. Acronyms DMN and IMA from Table 3 are not explained.

Relation to Prior Work: I am not that familiar with the field, but I think the work is novel. The related work section was an insightful read.

Reproducibility: Yes

Additional Feedback: In unification the same variable must be assigned the same value. Is this the case in this approach? I believe it’s not… How did you choose to use the same variable symbol in your figures?

[Author Response · NeurIPS 2020]

---

**Algorithm 1:** Unification Networks

---

**Input:** Invariant example $G$, variableness network $\psi$, example $K$, features network $\phi$, unifying features network $\phi_U$, upstream predictor network $f$

**Output:** Predicted label for example $K$

---

**begin** ▷ Unification Network
**return** $f \circ g(G, K)$                    ▷ Predictions using Soft Unification $g$

**begin** ▷ Soft Unification function $g$
**foreach** *symbol $s$* **in** $G$ **do**
$\boldsymbol{A}_{s,:} \leftarrow \phi(s)$                     ▷ Features of $G$, $\boldsymbol{A} \in \mathbb{R}^{|G| \times d}$
$\boldsymbol{B}_{s,:} \leftarrow \phi_U(s)$               ▷ Unifying features of $G$, $\boldsymbol{B} \in \mathbb{R}^{|G| \times d}$
**foreach** *symbol $s$* **in** $K$ **do**
$\boldsymbol{C}_{s,:} \leftarrow \phi(s)$                     ▷ Features of $K$, $\boldsymbol{C} \in \mathbb{R}^{|K| \times d}$
$\boldsymbol{D}_{s,:} \leftarrow \phi_U(s)$               ▷ Unifying features of $K$, $\boldsymbol{D} \in \mathbb{R}^{|K| \times d}$
**Let** $P = \text{softmax}(\boldsymbol{B}\boldsymbol{D}^T)$          ▷ Attention map over symbols, $\boldsymbol{P} \in \mathbb{R}^{|G| \times |K|}$
**Let** $\boldsymbol{E} = \boldsymbol{P}\boldsymbol{C}$             ▷ Attended representations of $G$, $\boldsymbol{E} \in \mathbb{R}^{|G| \times d}$
**foreach** *symbol $s$* **in** $G$ **do**
$\boldsymbol{U}_{s,:} \leftarrow \psi(s)\boldsymbol{E}_{s,:} + (1 - \psi(s))\boldsymbol{A}_{s,:}$    ▷ Unified representation of $G$ with $K$, $\boldsymbol{U} \in \mathbb{R}^{|G| \times d}$
**return** $\boldsymbol{U}$

---

Dear reviewers, thank you for your comments. We are pleased with the unanimous consensus on the novelty of our contribution, and recognition by some reviewers of a comprehensive evaluation. We thank Reviewer 3 for their suggestion of an algorithm box to clarify Section 2. We present Algorithm 1 above that defines our approach and the soft unification function $g$ with all the learnable components $\psi, \phi, \phi_U, f$. We compute 2 sets of features (see L87, L90, L refers to lines in paper), then use the unifying features to let each symbol in $G$ attend to a symbol in $K$. Depending on how much each symbol in $G$ is a variable, determined by $\psi$, the representation of $G$ is interpolated between its and $K$'s features. We plan to include the algorithm in Section 2 and simplify the notation, move Figure 2 to Section 3 and add further explanation using the extra page if the paper gets accepted. We hope these changes will resolve clarity issues also raised by Reviewers 1 and 4. Reviewer 3 asks about hyper-parameters and their selection as well as the number examples used for invariances. In Appendix A, second paragraph of each model details the dimensions, layers etc. used. These were selected based on similar previous works on the datasets. We use one example to learn from but have experimented with multiple (L209) of which the predictions are aggregated (e.g. sum), Appendix D Figure 8.

Reviewers 1 and 4 point out some baselines were missing in the text. We included them (e.g. DMN, G2N2) only as references in the captions due to space constraints; we will expand their definitions to clarify further. So in response to Reviewer 4, in Table 3 DMN and IMA are baselines without soft-unification and our approach achieves comparable results to DMN with half the data size. We explain weak supervision briefly in L213 and will clarify strong supervision further. To answer Reviewer 4, we do not constrain the same symbol variable to attain the same value because it can appear in different contexts. When visualising invariants, the choice of variable symbols is given by the variabless $\psi$ which is the same for same symbols. We leave contextualised variablisation as future work (L119). However, the value assignment can be context dependant (L123, L140).

Reviewer 2 points out there is no quantitative analysis of identifying invariants. This was not included because (i) the focus of our evaluation was to solve an upstream task with less examples using our approach whilst maintaining task performance, and (ii) our approach might solve the task without necessarily using the *expected* invariants as shown in Appendix D Figure 11 and 12. For completeness, we can add this analysis to our synthetic dataset where we know what are the expected invariances. Reviewer 2 mentions attention weights in Figure 5 are low $\leq 0.3$ (also mentioned by Reviewer 4) which might indicate low confidence. There might be a misunderstanding here as these are the variableness $\psi$ of the symbols, not the attention weights. We purposefully penalise the magnitude of $\psi$ in equation 3 (Sparsity) so we expect them to be low. This is because we want to find the minimum variablisation of G to correctly predict K's label (see L252). In Figures 1 and 6, we threshold $\psi$ for visualisation purposes (L227) although the interpretation of what symbols are variables is not binary, i.e. is it or is it not a variable. This is a soft view of the notion of a variable. Therefore, to answer Reviewer 2's comment on soft-interpolation, this soft view indeed produces varying symbol embeddings as intended. But this does not hurt the interpretability of the attention maps which are instead computed from the unifying features (Algorithm 1, line 10). The interpolation happens after the attention is computed.

Reviewer 2 asks about context embeddings to disamguiate unification. This is indeed used in some models through unifying features $\phi_U$ (L123, L140). In other words, in the sentence "$\boldsymbol{X}$:lily is a $\boldsymbol{Y}$:frog" what $\boldsymbol{X}$ unifies with takes into account that it is related to a frog / animal, e.g. Figure 3 pink diamond represents unification RNN (L140). We thank Reviewer 2 for the suggested related work [a] and will cite it; however, we do not claim our approach is more interpretable than Neural Turing Machines (NTM) (L273). We discuss NTMs as an approach that lacks an explicit representation like our invariants (L278) in relation to generalisation. We acknowledge that interpretability can be subjective and biased (L281), for example we might want to think $\boldsymbol{X}$:Mary means *someone* but we do not know how the model uses these representations (L297, L300). We discuss why and why not our results could be interpretable but refrain from claiming that our model provides an explanation like Grad-CAM (L301) or is more interpretable than NTMs.

We thank you for your reviews and hope to have addressed your questions. We intend to incorporate all the suggestions and remaining remarks into the paper.

[Meta-Review · NeurIPS 2020]

Pros: - Topic is interesting: learning something akin to symbolic reasoning - Nice idea of Identifying invariants through soft-attention in expressions via task- specific examples - Informative rebuttal addressing many of the points raise - Paper well written Cons: - Missing some analysis to understand what the model is doing (show with ground truth the success in identifying variables) After reading the author’s response, R3 decided to increase his/her score to marginally above acceptance threshold, all other reviewers maintained their original scores. R3 finds the author’s response satisfactory, but would like to see analysis over learning rates. In particular, he/she considers it useful to include training curves. R2 recommends to reject the paper considering that the authors should provide a quantitative-ablative analysis of the various components in the framework to substantiate their claims. And considers it to be essential to show how well the model can correctly identify the variables (at least in the synthetic case). While R4 agrees with the points raised by R3 and R2, he/she argues that the novelty of the approach (and relevance of the problem) renders the paper particularly interesting. The AC agrees with this view. The AC considers the current results informative and given that the authors promised to report a quantitative analysis of identifying invariants, recommends accepting the paper. The AC also encourages the authors to report training curves as mentioned by R3.